# Elucidation of Agonist and Antagonist Dynamic Binding Patterns in ER-α by Integration of Molecular Docking, Molecular Dynamics Simulations and Quantum Mechanical Calculations

**DOI:** 10.3390/ijms22179371

**Published:** 2021-08-29

**Authors:** Sugunadevi Sakkiah, Chandrabose Selvaraj, Wenjing Guo, Jie Liu, Weigong Ge, Tucker A. Patterson, Huixiao Hong

**Affiliations:** Division of Bioinformatics and Biostatistics, National Center for Toxicological Research, U.S. Food and Drug Administration, 3900 NCTR Road, Jefferson, AR 72079, USA; Suguna.Sakkiah@fda.hhs.gov (S.S.); selnikraj@bioclues.org (C.S.); Wenjing.Guo@fda.hhs.gov (W.G.); Jie.Liu1@fda.hhs.gov (J.L.); Weigong.Ge@fda.hhs.gov (W.G.); Tucker.Patterson@fda.hhs.gov (T.A.P.)

**Keywords:** molecular docking, molecular dynamics simulations, quantum mechanical calculations, estrogen receptor, dynamic binding pattern

## Abstract

Estrogen receptor alpha (ERα) is a ligand-dependent transcriptional factor in the nuclear receptor superfamily. Many structures of ERα bound with agonists and antagonists have been determined. However, the dynamic binding patterns of agonists and antagonists in the binding site of ERα remains unclear. Therefore, we performed molecular docking, molecular dynamics (MD) simulations, and quantum mechanical calculations to elucidate agonist and antagonist dynamic binding patterns in ERα. 17β-estradiol (E2) and 4-hydroxytamoxifen (OHT) were docked in the ligand binding pockets of the agonist and antagonist bound ERα. The best complex conformations from molecular docking were subjected to 100 nanosecond MD simulations. Hierarchical clustering was conducted to group the structures in the trajectory from MD simulations. The representative structure from each cluster was selected to calculate the binding interaction energy value for elucidation of the dynamic binding patterns of agonists and antagonists in the binding site of ERα. The binding interaction energy analysis revealed that OHT binds ERα more tightly in the antagonist conformer, while E2 prefers the agonist conformer. The results may help identify ERα antagonists as drug candidates and facilitate risk assessment of chemicals through ER-mediated responses.

## 1. Introduction

Estrogen receptor (ER) is one of the important targets of drugs and endocrine disrupting chemicals in the endocrine system [1]. It is a ligand-dependent transcriptional factor in the steroid type 1 nuclear receptor family [2]. ER plays a major role in various biological functions such as bone modeling, reproductive system, cardiovascular system, metabolism, and cell proliferation [3]. ER is an extensively studied target among the endocrine receptors. There are two major ER isoforms, ERα and ERβ. Like other nuclear receptors, ERα consists of three distinct domains: N-terminal domain (residue 1–180), DNA binding domain (residue 181–263), and C-terminal domain or ligand binding domain (LBD, residue 303–552) (Figure 1). The activation function domain 1 (AF1) is present in the N-terminal domain and plays a major role in the protein–protein interaction [4,5]. The mitogen-activated protein (MAP) kinase pathway regulates the activity of AF1 through the growth factors [6]. The LBD is composed of twelve helices and two antiparallel β-sheets which are arranged as a three-layer antiparallel α helical sandwich [5,7]. The first layer is formed by helices 1 to 4 and 7, the middle layer is made up of helices 5, 6, 9 and 10 and the final layer is composed of helices 8, and 11 [8,9,10]. The activation function domain 2 (AF2) in LBD is responsible for binding of cofactors. AF2 undergoes conformational change due to the binding of a compound in the ligand binding pocket (LBP). The conformational change of AF2 determines the types of binding cofactors which play a major role in activating or inhibiting the target genes of ER [11]. The hinge (residue 264–302) regions connect the DBD and LBD.

The H12 acts as a molecular switch that turns ER activity on and off depending on the binding chemicals [11,12,13]. The AF1 and AF2 play a major role in the transcriptional activation of ER [14]. The estrogenic compounds bind in the hydrophobic pocket of ER LBD. The hydrophobic pocket is composed of Met342 to Leu354 of H3, Trp383 to Arg394 of H6, Val418 to Leu428 from the preceding loop of H8, Met517 to Met528 of H11, Leu539 to His547 of H12, and Leu402 to Leu 410 of S1/S2 hairpin [7]. Binding of antiestrogenic compounds to ER induce a H12 conformational changes by placing H12 across the H3 and H11 and moving H12 away from the LBP. Due to the H12 conformation change, the AF2 in the LBD is distorted and not suitable for binding cofactors [15]. ER enhances and represses its function via various pathways [10,16,17]. Understanding the ERα dynamic binding patterns with agonists and antagonists is crucial for discovery of ERα agonists and antagonists. Dynamic binding pattern represents the forming and breaking of non-covalent interactions such as hydrogen bonding and Van der Waals interactions between a protein and a ligand, as well as conformational changes caused by the binding ligand throughout a molecular dynamics (MD) simulation. More than 350 3D structures of ER bound with various ligands are deposited in the Protein Data Bank (PDB). Those structures are useful to understand the structural changes due to agonist and antagonist binding in the ERα LBP. Various computational techniques such as molecular docking [18,19,20,21,22,23,24], MD simulations [25,26,27,28,29,30], predictive modeling [31,32,33,34,35,36,37,38,39,40,41], and in vitro studies were conducted to predict ER binders or non-binders [42,43] and agonists or antagonists [44,45].

Many ligand-based computational methods were used to predict ERα activity of chemicals based on chemical features, including ERα binders and nonbinders, and agonist and antagonist activities [31,46,47,48,49]. However, the dynamic binding patterns of ER agonists and antagonists are not clearly understood. Hence, in this study we applied QM-Polarized Ligand Docking (QPLD) and MD simulations to elucidate the dynamic binding patterns of ERα agonists and antagonists using 17β-estradiol (E2) and 4-hydroxytamoxifen (OHT). E2 is the natural steroid hormone that activates ER. The activated ER modulates gene expression in cells. E2 binds with ER in the nucleus and forms a dimer. Subsequently, the dimer interacts with the estrogen response element of ER and regulates transcription of the target gene. E2 has two hydroxyl groups, one at C3 and another at 17β (Figure 2). The hydroxyl group at C3 forms hydrogen bonds with Glu353 and Arg394 of ER. The 17β-OH group forms a hydrogen bond with His524. The planar part of A/B ring forms a sandwich between Ala350 and Leu387. The D ring forms a nonpolar contact with Ile424, Gly521 and Leu525 [7]. OHT is a selective estrogen receptor modulator and acts as an antagonist towards ER in specific tissues [6]. The hydroxyl group in OHT (Figure 2) has a high binding affinity towards ER [50]. Binding of OHT in the LBP of ER pushes the H12 away to occupy part of the AF2 site, blocking coactivators binding in AF2 [51]. In QPLD, the ab initio molecular charges were applied to obtain binding orientation of the two chemicals in the LBP of ERα [52]. MD simulations were used to elucidate the dynamic binding patterns of the agonist E2 and antagonist OHT.

## 2. Results

### 2.1. Molecular Docking

QPLD and Glide docking are two widely used docking methods to identify orientations of compounds in binding sites of proteins [52,53,54]. The EXtra-Precision (XP) Glide scores, QPLD scores and docking energy values of the four complexes with the best ligand orientations are shown in Table 1.

In both docking methods, the agonist E2 had a lower docking energy in the agonist conformation (ERα1) than in the antagonist conformation (ERα2), while the antagonist OHT had a higher docking energy in the agonist conformation (ERα1) than in the antagonist conformation (ERα2). The orientations of E2 and OHT in ERα1 and ERα2, as well as the ERα residues interacting with E2 and OHT in the four complexes, are depicted in Figure 3. E2 in the binding site of ERα1 forms interactions with Glu353, Arg394, and His524. E2 also forms interactions with Glu353 and Arg394 but fails to interact with His524 in the binding site of ERα2. OHT forms hydrogen bond interactions with both Arg344 and Glu353 in the binding site of ERα2. OHT interacts with Arg344 but fails to interact with Glu353 in ERα1. The interaction analysis revealed that E2 forms more hydrogen bond interactions in ERα1, while OHT forms more hydrogen bond interactions in ERα2. These four complexes were subjected to MD simulations to elucidate the dynamic binding patterns of agonist E2 and antagonist OHT in ERα.

### 2.2. MD Simulations

In the MD simulations for each of the four complex structures obtained from molecular docking, structures were recorded for every 4.8 picoseconds (ps) in the trajectory file. Thus, each trajectory file contains 20,835 structures. The details of the simulation systems are summarized in Table 2.

To understand the dynamics of ER binding with the agonist E2 and antagonist OHT, root mean square deviations (RMSD) were calculated between the 20,835 structures for ERα1_E2, ERα1_OHT, ERα2_E2, and ERα2_OHT using a MATLAB script. The obtained RMSD matrixes for the four complexes are shown in Figure 4. Examining the RMSD values from the MD simulations (shown in Figure 4) found that structural changes were not the same during the MD simulations and the RMSD matrixes formed patterns. Furthermore, the RMSD patterns are different among the four complexes, indicating the binding dynamics of the agonist and antagonist in ERα are different.

The interaction energy was calculated using prime MM/GBSA for whole trajectory files and the calculated energy values are provided in Table 3.

To identify distinct structural patterns, hierarchical clustering analysis was conducted based on the RMSD matrixes. The major clusters (with >500 structures) from the MD simulations for ERα1_E2, ERα1_OHT, ERα2_E2, and ERα2_OHT are summarized in Table 4.

To further examine conformational changes of ER caused by agonist (E2) and antagonist (OHT) binding in the simulations, we used the distance between H12 and the centroid of LBP of ER to measure the conformational changes of H12. The residues (only in the secondary structure, not from loop region) around 4 Å to the ligands in 1GWR and 3ERT were selected for calculating the centroid of LBP: Met343, Leu346, Thr347, Leu349, Ala350, Asp351, Glu353, Leu384, Leu387, Leu391, Arg394, Phe404, Met421, Leu428, Gly521, His524 and Leu525. The residues Asp535 to Leu549 were selected to represent H12. First, a distance was calculated between the centroid and each of the H12 residues for a structure in the simulations. The maximum of the H12 distances was used to measure the distance of H12 from the centroid. For each structure in the simulations, a relative distance was calculated by subtracting the distance of the initial structure. The resulting relative distances for simulations of ERα2_E2 and ERα1_OHT are shown in Figure 5. Most of the structures in the simulation of ERα1-OHT had a longer distance between H12 and the centroid of LBP than the initial structure (positive relative distances, top of Figure 5), indicating the antagonist OHT made conformational changes in H12 from the initial active form towards to inactive form. On the other hand, most of the frames in the trajectory file from simulation of ERα2_E2 had a shorter distance between H12 and the centroid of LBP than the initial structure (negative relative distances, bottom of Figure 5), indicating the agonist E2 caused conformational changes in H12 from the initial inactive form towards the active form. Then, based on the determination of the active and inactive forms using the relative H12 distances, the receptor activation energies were calculated using the standard thermodynamic relation ΔG = −RTln(N_active_/N_inactive_) [55], resulting 1.48 Kcal/mol for ERα1_OHT and 3.51 Kcal/mol for ERα2-E2.

### 2.3. Energy Analysis

A molecular mechanics-generalized Born/surface area (MM-GB/SA) approach was used to calculate the binding free energies for ERα1_E2, ERα2_E2, ERα1_OHT, and ERα2_OHT complexes. The binding free energy for the representative structure from each cluster of ERα1_E2, ERα1_OHT, ERα2_E2, and ERα2_OHT complexes are depicted in Figure 6. The binding free energy analysis plot shows different patterns of energy travel for complexes.

For ERα1_E2 and ERα2_E2, the free energy values shifted from −86.70 to −97.43 Kcal/mol and from −92.87 to −96.03 Kcal/mol in the MD simulations, respectively (top of Figure 6). In the early simulation time, a higher free energy value −86.70 Kcal/mol was observed for the complex ERα1_E2 than for the complex ERα2_E2 which had a free energy value of −92.87 Kcal/mol. At the end of the simulations, the complex ERα1_E2 had a lower free energy value (−97.43 Kcal/mol) than the complex ERα2_E2 (−96.03 Kcal/mol). The dynamic patterns of free energy revealed that E2 can accommodate better in the LBP of the ER agonist conformer than the ER antagonist conformer.

For ERα1_OHT and ERα2_OHT, the free energy travels from −134.20 to −123.20 Kcal/mol and from −106.31 to −101.07 Kcal/mol, respectively (bottom of Figure 6). The increase in free energy of ERα2_OHT in the simulation was smaller compared to ERα1_OHT. The larger increase in free energy for ERα1_OHT in the simulation might be due to H12 changing from an agonist conformation to an antagonist conformation. The analysis of free energy pattern in the simulations indicated that OHT had stronger binding than E2 in the hydrophobic LBP of ERα.

The dynamic binding interaction pattern analysis revealed that E2 and OHT bind tightly in the LBP of ERα1 and ERα2. Throughout the simulation, 12 residues of ERα1 interacted with E2. Among the 12 residues, His524 and Glu353 had hydrogen bond interactions in more than 60% and less than 20% of the simulation time, respectively. Phe404 had hydrophobic interactions in more than 60% of the simulation time. Though Ala350, Leu384, Met388, Leu391, Ile424 and Leu525 had hydrophobic interactions, the interactions were observed in less than 20% of the simulation time (Figure 7). The hydrogen bond between E2 and His524 and the hydrophobic interaction between E2 and Phe404 were the most stable in ERα1_E2 complex, while the other observed interactions were transient. Thirty-three residues of ERα2 formed interactions with OHT. Among the 33 residues, Glu353 had hydrogen bond interactions with OHT in more than 70% of the simulation time, while Thr347, Arg394 and Phe404 had hydrogen bond interactions with OHT in less than 20% of the simulation time. Ala350 and Leu525 had hydrophobic interactions with OHT in more than 40% of the simulation time, while Met343, Leu346, Leu354, Trp383, Leu384, Leu387, Met388, Leu391, Phe404, Met421, Ile424, Leu428 and His525 had hydrophobic interactions with OHT in less than 30% of the simulation time (Figure 7). Therefore, the hydrogen bond between OHT and Glu353 and hydrophobic interactions between OHT and Ala350 and Leu525 were more stable than other observed interactions. The dynamic binding interactions pattern analysis revealed that OHT tightly binds ERα2.

## 3. Discussion

ERα is one of the well-studied targets in the endocrine system. It plays a major role in various diseases such as cancer, bone modeling, reproductive system, cardiovascular system, metabolism, and cell proliferation [3]. Until now, many estrogenic activity chemicals were identified and crystallized in the binding site of the ER. Various computational and experimental studies were carried out to predict estrogenic activity of chemicals such as agonist, antagonist, binder, or non-binder. However, the ERα dynamic binding patterns for agonist and antagonist remain unclear. Hence, in this study, various computational techniques were applied to gain insight to the dynamic binding patterns of agonist and antagonist in the binding site of ERα. The 3D structure of ER complexed with E2 and OHT were retrieved from the PDB. The ERα1 and ERα2 represents the ER conformation in presence of E2 and OHT, respectively. The ERα1 and ERα2 were used as the target proteins to dock the E2 and OHT using QPLD.

E2 and OHT were redocked in the binding sites of ERα1 and ERα2, respectively, to validate the docking procedure and parameters [56]. The complexes from redocking were superimposed with the X-ray crystal structures 1GWR and 3ERT, respectively, to calculate RMSD values for the ligands (Figure 8). The RMSD values for E2 and OHT were 0.309 Å and 0.439 Å, respectively. The small RMSD values indicate that the docking procedure used and parameters are reliable.

The XP scores of E2 in the LBP of ERα1 and ERα2 were −11.00 and −9.65 Kcal/mol, respectively. The ERα1 and ERα2 complex with OHT had XP scores −9.07 and −8.59 Kcal/mol, respectively. Based on the Glide XP scores, E2 and OHT can bind tightly in the binding site of ERα1 compared to ERα2. The analysis based on the QPLD scores showed a different result compared to the Glide XP scores. Based on the QPLD scores, E2 and OHT can tightly bind in the binding site of ERα1 and ERα2, respectively. Applying ab initio charges can significantly increase the predictive power of the orientation of the compounds in the binding site of a protein. Hence, the ERα complex with E2 and OHT were selected based on the QPLD scores for interaction analysis instead of the Glide XP docking scores. The selected four ER complexes (ERα1_E2, ERα1_OHT, ERα2_E2, and ERα2_OHT) were subjected to MD simulations.

The in-house MATLAB script was used to calculate the RMSD matrixes from MD simulation trajectory files of the ERα complexes. The heat maps were generated based on the RMSD matrixes to identify structure clusters in the trajectory files (Figure 4). The heat maps showed various patterns of structural changes in the MD simulations of the four complexes. Hence, the hierarchical clustering method was used to group similar conformations of the ER complex from each trajectory based on its RMSD values. The clustering analysis revealed 6, 4, 5, and 4 clusters for ERα1_E2, ERα1_OHT, ERα2_OHT, and ERα2_E2, respectively. The binding free energy value was calculated for the representative structures from each cluster. For ERα1_E2 and ERα2_E2, the free energy values shifted from −86.70 to −97.43 Kcal/mol and −92.87 to −96.03 Kcal/mol, respectively. For ERα1_OHT and ERα2_OHT, the free energy travels from −134.20 to −123.20 Kcal/mol and from −106.31 to −101.07 Kcal/mol, respectively. The free energy analysis revealed that OHT had tighter binding than E2 in the hydrophobic ligand pocket of ERα1 and ERα2. So, to dissociate OHT from ERα higher external energy is required than for E2. Thus, the antagonist stays longer in the LBP of ER and represses its function.

## 4. Materials and Methods

### 4.1. Study Design

The overall workflow of this study is depicted in Figure 9. The ER complexed with E2 (PDB ID:1GWR, agonist) and OHT (PDB ID:3ERT, antagonist) were downloaded from the PDB. ER in the agonist and antagonist conformations was named ERα1 and ERα2, respectively. The QPLD method was applied to re- and cross-dock E2 and OHT in the LBP of ERα1 and ERα2 conformation. Four ER complexes (ERα1_E2, ERα1_OHT, ERα2_E2, and ERα2_OHT) were obtained from the molecular docking. The four ER complexes were subjected to 100-nanosecond (ns) MD simulations using DESMOND (https://www.schrodinger.com/products/desmond, (accessed on 8 August 2021)-Maestro-Desmond v-44017 Interoperability Tools, Schrödinger, New York, NY, USA). A MATLAB script was written to generate a RMSD matrix from each trajectory file. The hierarchical clustering analysis was carried out based on RMSD values and a representative structure was selected from each cluster. The binding free energy value was calculated by prime MM-GB/SA for each representative structure of the ER complexes from clustering analysis.

### 4.2. Molecular Docking

#### 4.2.1. Protein Preparation WIZARD

The E2 complex structures, 1GWR and 3ERT, were downloaded from the PDB. 1GWR is a dimer form of ER complexed with E2. In this study, only the chain A with E2 was selected. Protein Preparation Wizard from Maestro in Schrodinger suite (https://www.schrodinger.com/products/protein-preparation-wizard, (accessed on 8 August 2021) version 2015-4 Schrödinger, LLC, New York, NY, USA) was used to re-move the heteroatoms and water molecules which are not interacting with binding site residues from 1GWR_A (ERα1) and 3ERT (ERα2). The Prime module in Schrodinger suite (https://www.schrodinger.com/products/prime, (accessed on 8 August 2021) version 2015-4 Schrödinger, LLC, New York, NY, USA) was used to add proper hydrogen atoms and to build the missing atom or residues in ER. Subsequently, the structures were optimized and minimized by applying the OPLS-AA-2005 force field [57]. The minimized ERα1 and ERα2 structures were used as the receptors to dock E2 and OHT. A 2 Å grid box was generated around the E2 and OHT in the binding site of ERα1 and ERα2, respectively.

#### 4.2.2. Ligand Preparation

E2 and OHT were extracted from the structures 1GWR and 3ERT, respectively. These two compounds were imported into the LigPrep module in Schrodinger suite (https://www.schrodinger.com/products/ligprep, (accessed on 8 August 2021) version 2015-4 Schrödinger, LLC, New York, NY, USA) to check the bond order, ionization states, steric isomers and search the tautomers. The conformers were generated using the ConfGen method in the LigPrep module with distance dependent dielectric solvation and OPLS-AA_2005 force field used for energy minimization. The prepared E2 and OHT were saved in the SDF format.

#### 4.2.3. QPLD Molecular Docking

QPLD is one of the powerful docking methods to identify the best orientation of a chemical in the binding site of a protein. As illustrated in Figure 10, QPLD combines the Glide docking algorithm with QM/MM calculations by Q-site program which uses the Jaguar and Impact program for the ligand with binding site residues (QM) and the remaining regions of the protein (MM), respectively. Subsequently, the ab initio charge was applied for the chemicals calculated using the Q-site which uses the 6-31G**/LACVP* basis set, B3LYP and Ultrafine SCF accuracy level for the density function calculation.

Initially, the Glide molecular docking module was used to dock E2 and OHT in the LBP of ERα1 and ERα2 using XP mode. GRID files were prepared by selecting residues 5 Å around the ligands. QM-based charge generated by DFT method was incorporated. The poses generated from Glide docking for the four complex structures (ERα1_E2, ERα1_OHT, ERα2_E2, and ERα2_OHT) by XP mode were subjected to QPLD to redock E2 and OHT in the LBP of ER1α and ER2α. First, QPLD generates several unique ER complexes with E2 and OHT. The QM regions were assigned for the ligand and the residues within 5 Å to the ligands (Figure 11).

The generated ER complexes were automatically subjected to the Q-site to compute the single point energy for each ER complex. Once the charge was calculated, E2 and OHT were redocked in the LBP of ERα1 and ERα2 by Glide module. Finally, 20 different complexes with XP score, docking energy, and the QPLD score were calculated.

### 4.3. MD Simulations

MD simulations were used to optimize the conformation of the docked ER complexes by analyzing the movements of atoms. The best conformation of ERα1_E2, ERα1_OHT, ERα2_E2, and ERα2_OHT from QPLD were subjected to 100 ns MD simulations using DESMOND (https://deshawresearch.com, (accessed on 8 August 2021) Maestro-Desmond v-44017 Interoperability Tools, Schrödinger, New York, NY, USA). The simulation systems were prepared by applying the OPLS-AA-2005 force field to the ER and ligands (E2 and OHT) and an orthorhombic box was generated by 10 Å border from the ER complexes. The accuracy of MD simulation is impacted by the force field used for the components in a simulation system. A specific force field should be derived and used for a nonstandard molecule [58]. The OPLS parameters are optimized for a variety of structural features in small molecules such as E2 and OHT to reproduce the thermodynamic properties in the liquid state [59,60,61]. The orthorhombic box was filled with TIP3P water molecules. To neutralize the whole simulation system, 0.15 M Na^+^ and Cl^−^ were added based on the total charge of the ER complex. The ER simulation systems were subjected to two-step energy minimizations with and without the restraint applied on the solute. As a first step of energy minimization, the ER complex was subjected to 12 ps of NVT simulation carried out at 10 K with Berendsen barostat. A restraint was applied to all the heavy atoms of the ER complexes, followed by 12 ps of NPT simulations carried out at 1 atmospheric pressure and 10 K with Berendsen barostat. The temperature and pressure were kept constant at 310 K for 100 ns. Further, the relaxed systems were subjected to a 100-ns simulation with a time set of two femtoseconds. The final trajectory files were saved for every 4.8 ps and used for subsequent analysis.

The trajectory files were used to elucidate the ER molecular mechanism in the presence of E2 and OHT. Initially, the RMSD matrices were created using the MATLAB script. The structures in each trajectory file were clustered based on the RMSD values calculated between them. The clusters with more than 500 structures were selected. A representative structure was obtained from each cluster for energy analysis.

### 4.4. Energy Calculation

The representative structures of ERα1 and ERα2 complexes from the clustering analysis were used to calculate the binding free energy. The MM-GB/SA method and OPLS-AA_2005 force field were used to compute the electrostatic component of the solvation free energy. This approach combines the molecular mechanism and continuum solvent models to predict the protein–ligand binding free energy as illustrated by the thermodynamic cycle shown in in Figure 12.

First, the receptor (ERα1 or ERα2), ligand (E2 or OHT), and complex of a receptor bound by a ligand were optimized in the solvent environment. The optimized structures were then used to calculate energy terms such as coulomb energy, covalent binding energy, Van der Waals energy, lipophilic energy, generalized Born electrostatic solvation energy, hydrogen bonding correction, and pi-pi stacking correction. These energy terms for the receptor, ligand, and complex were summed up as free energy as shown in equations below.
$$E_{Protein\;}=\;E_{Protein}^{Coulomb}+E_{Protein\;}^{Covalent}+E_{Protein}^{Hbond}+E_{Protein}^{Lipo}+E_{Protein}^{Solv\_GB}+E_{Protein}^{vdW}+E_{Protein}^{Packing}$$
$$E_{Ligand\;}=\;E_{Ligand}^{Coulomb}+E_{Ligand\;}^{Covalent}+E_{Ligand}^{Hbond}+E_{Ligand}^{Lipo}+E_{Ligand}^{Solv\_GB}+E_{Ligand}^{vdW}+E_{Protein}^{Packing}$$
$$E_{Complex\;}=\;E_{Complex}^{Coulomb}+E_{complex\;}^{Covalent}+E_{Complex}^{Hbond}+E_{Complex}^{Lipo}+E_{Complex}^{Solv\_GB}+E_{Complex}^{vdW}+E_{Complex}^{Packing}$$

*E_Coulomb_, E_Covalent_*, *E_Hbond_*, *E_Lipo_, E_Solv_GB_*, *E_vdW_*, and *E_Packing_* represent coulomb energy, covalent binding energy, hydrogen bonding correction, lipophilic energy, generalized born electrostatic solvation energy, Van der Waals energy, and pi-pi packing correction, respectively. The binding free energy is estimated from the free energies of the ligand, protein and the complex using the following equations.
Δ*G_bind_* = *E_complex_* − *E_protein_* − *E_ligand_*

Δ*G_bind_* is the estimated binding free energy. *E_complex_* is the estimated free energy of the complex in solvent, *E_Protein_* is the free energy of the protein in solvent, and *E_ligand_* is the free energy of the ligand in solvent.

## 5. Conclusions

ER is one of the important endocrine targets in the endocrine system. Overexpression of ER leads to various diseases such as cancer. Hence, understanding the dynamic binding patterns of agonist and antagonist binding in the hydrophobic binding pocket of ER gives insight into designing and predicting effective estrogenic compounds. Here, we applied an approach combining QM/MM docking and MD simulations to determine the energy-based ER agonist and antagonist binding mechanisms. The molecular docking revealed that E2 and OHT tightly bind in the LBP of ERα1 and ERα2, respectively. QPLD more accurately predicted the binding orientation of the estrogenic compounds in the LBP of ER than Glide docking. The heat maps RMSD values revealed that different clusters formed for the structures during the MD simulations and the binding mechanisms of agonist and antagonist in ERα were different. The binding free energy analysis for the representative structures of the clusters revealed that OHT binds more tightly with ERα2, while E2 prefers to bind ERα1. The binding interaction analysis revealed that the hydrogen bond between OHT and Glu353 and the hydrophobic interaction between OHT and Ala350 and Leu525 are the most stable interactions in the ERα2_OHT complex. These interactions might be the reason for the tight binding of OHT rather than E2 in the LBP of ERα1 and ERα2. Our findings shed light on the structural basis of agonist and antagonist dynamic binding pattern. This insight into the binding pattern and free energy analysis may help to discover ERα agonists and antagonists as drug candidates and facilitate risk assessment of chemicals through ER-mediated responses.

## Figures and Tables

**Figure 1 ijms-22-09371-f001:**
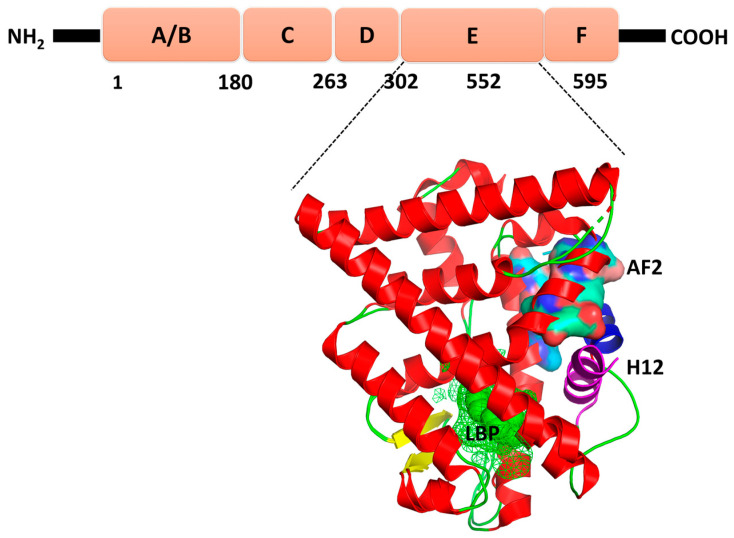
The cartoon representation of the different domains of ER. LBP is shown in green; AF2 is shown in electrostatic representation; and H12 is shown in magenta and blue for active (agonist bound) and inactive (antagonist bound) forms, respectively.

**Figure 2 ijms-22-09371-f002:**
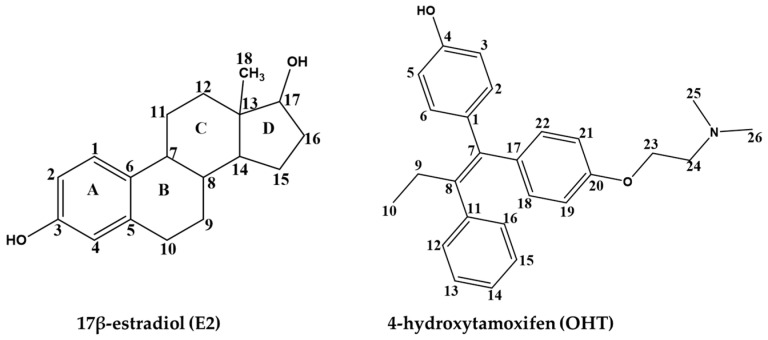
Two-dimensional structures of 17β-estradiol (E2) and 4-hydroxytamoxifen (OHT).

**Figure 3 ijms-22-09371-f003:**
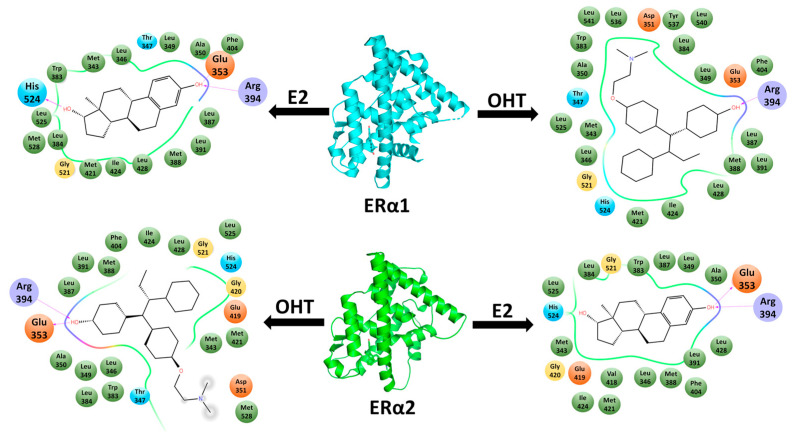
Binding orientations of E2 and OHT in the hydrophobic binding pocket of ERα1 and ERα2. The dotted lines represent hydrogen bond interactions. Green circles represent the hydrophobic residues; cyan circles represent the polar residues; red and blue ovals represent the negative and positive charged residues, respectively.

**Figure 4 ijms-22-09371-f004:**
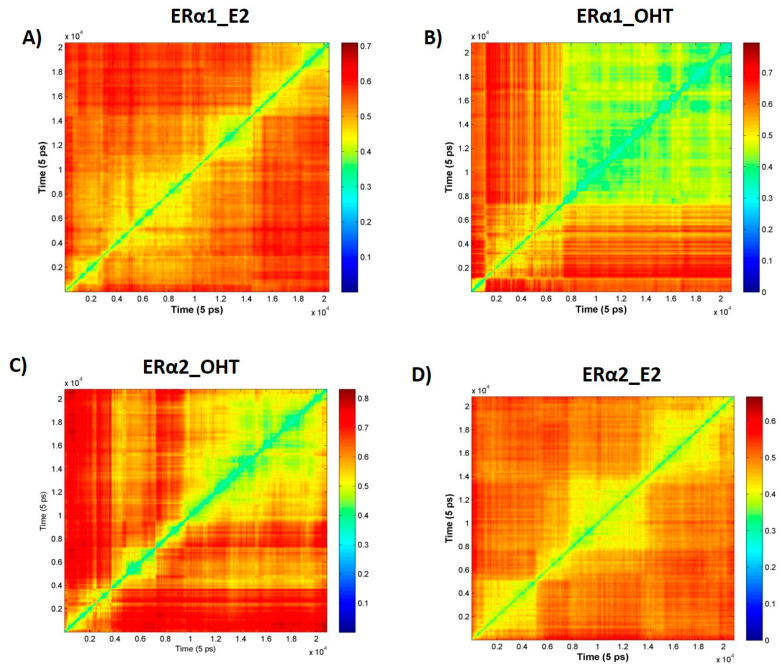
RMSD matrixes from MD simulations of the four complexes: (**A**) ERα1_E2, (**B**) ERα1_OHT, (**C**) ERα2_OHT, and (**D**) ERα2_E2. The complex is shown in the title of each panel. The time of a structure in the MD simulations is depicted by the axis. The RMSD values are color coded as shown in the subfigure color legends.

**Figure 5 ijms-22-09371-f005:**
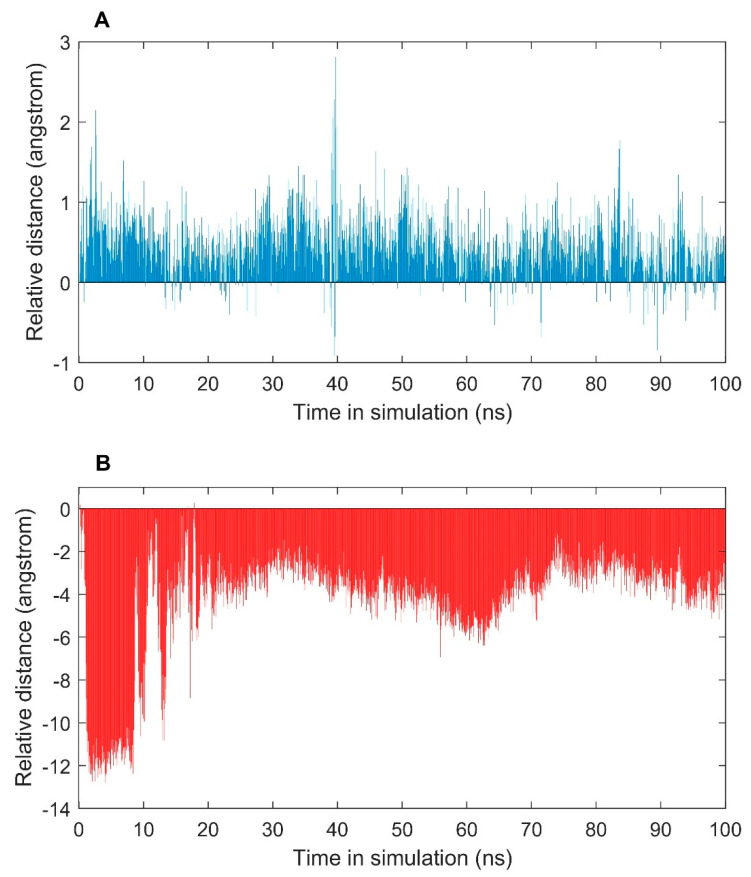
Relative distances of H12 to the centroid of LBP of ER in the simulations of ERα1_OHT (**A**) and ERα2_E2 (**B**). The X-axis indicates simulation time in ns. The Y-axis shows the relative H12 distance compared to the initial structures.

**Figure 6 ijms-22-09371-f006:**
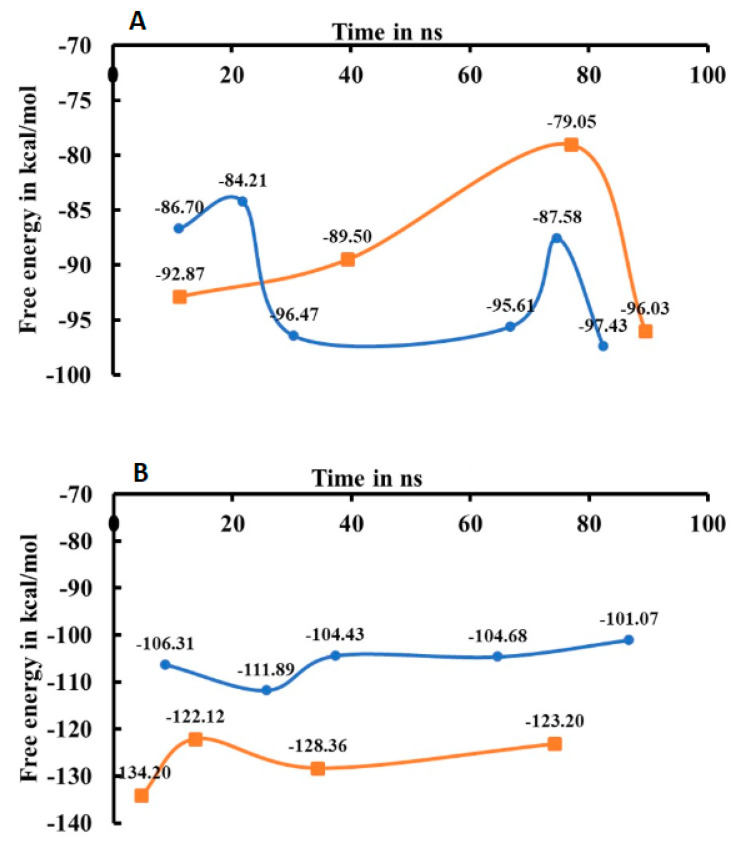
Binding free energy analysis on the average structure from each cluster of ER complexes ERα2_E2 (brown in **A**), ERα1_E2 (blue in **A**), ERα1_OHT (brown in **B**), and ERα2_OHT (blue in **B**) calculated using Prime MM-GB/SA.

**Figure 7 ijms-22-09371-f007:**
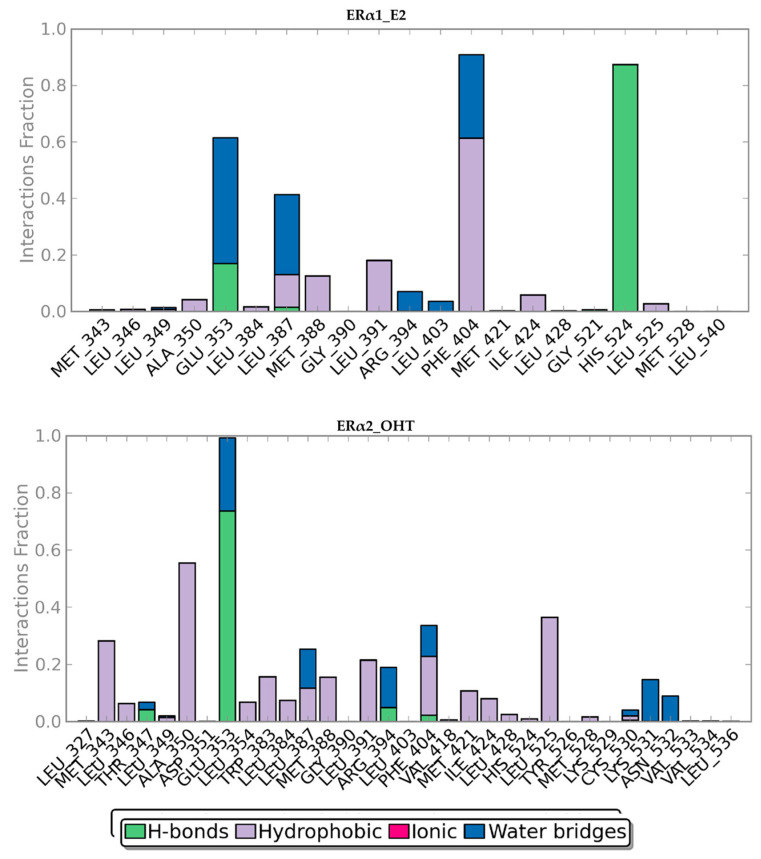
Interactions between E2 and ERα1 (**top**) and between OHT and ERα2 (**bottom**). The X-axis represents amino acids and their numbering in ER. The Y-axis shows the fractions of simulation time for different interaction types which marked in different colors.

**Figure 8 ijms-22-09371-f008:**
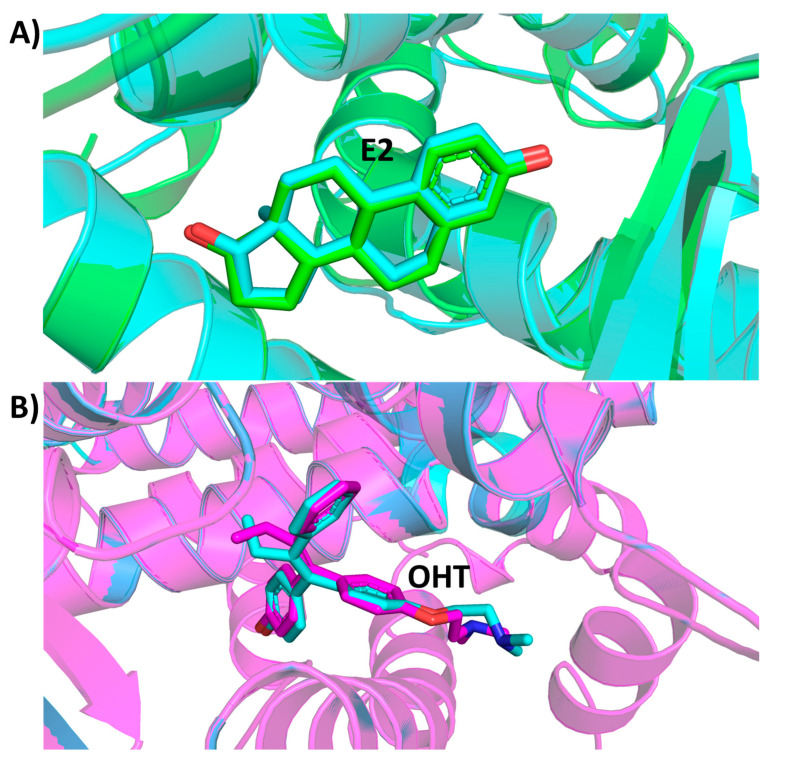
The superimposition of the redocked and X-ray crystal structure of ER complexes. (**A**) Docked ERα1_E2 complex and X-ray crystal structure (1GWR). (**B**) Docked ERα2_OHT complex and X-ray crystal structure (3ERT). The ER are shown in the stylized model. E2 and OHT are displayed in the stick model. X-ray structures are in cyan, docked ERα1_E2 in green and ERα2_OHT in magenta.

**Figure 9 ijms-22-09371-f009:**
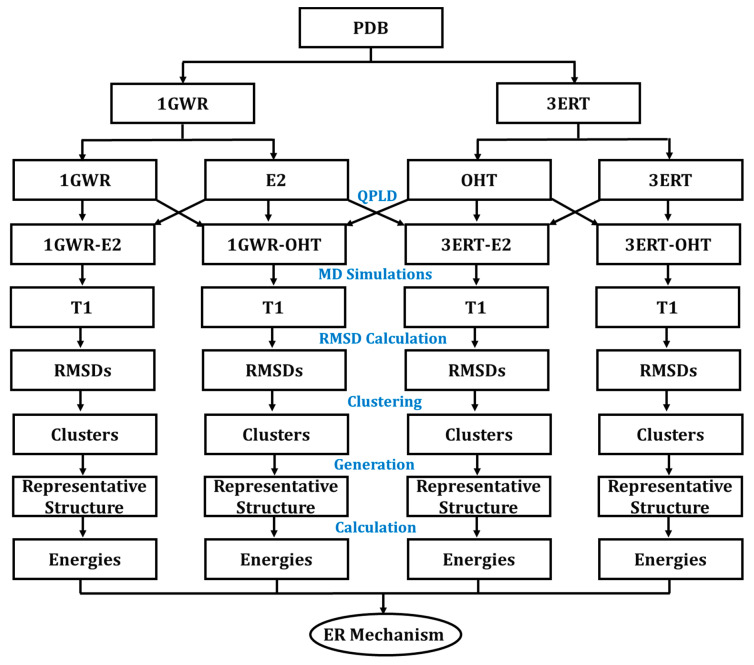
Overview of the study design.

**Figure 10 ijms-22-09371-f010:**
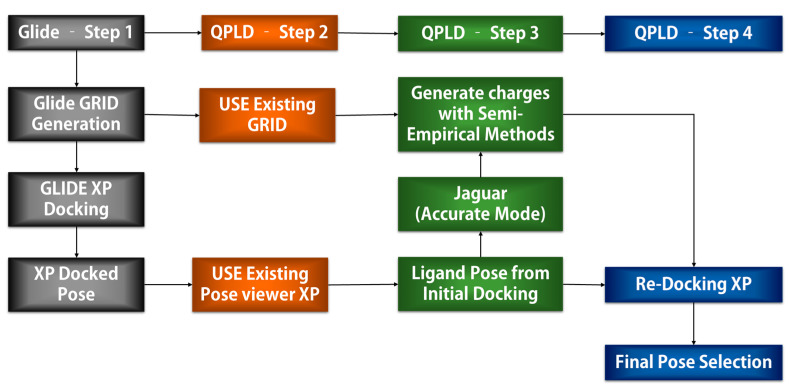
QPLD docking procedure.

**Figure 11 ijms-22-09371-f011:**
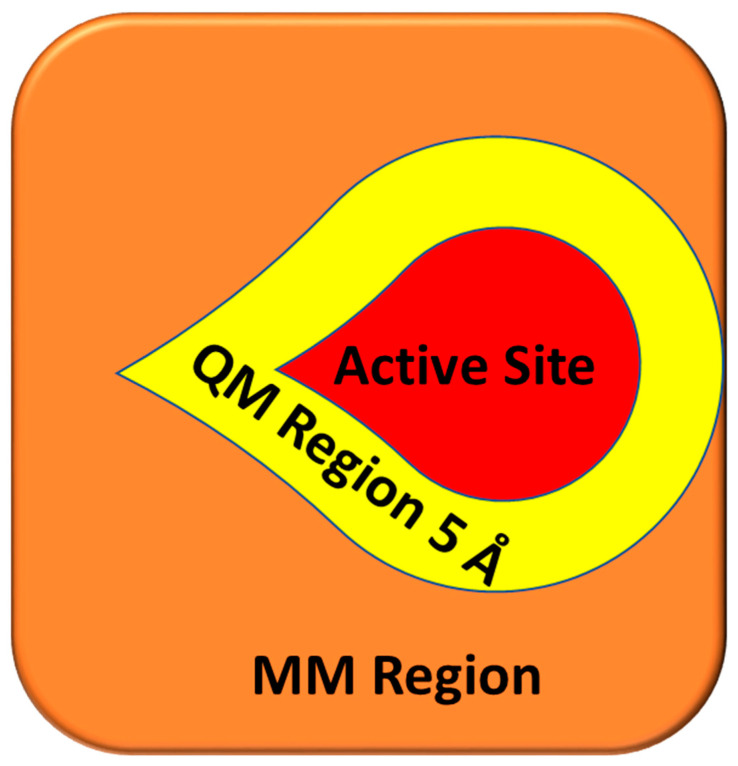
Cartoon representation of the region assigned for QM and MM during the QPLD docking. The square represents ER, binding site is highlighted in red, the QM region is depicted in yellow, and the MM region is shown in orange.

**Figure 12 ijms-22-09371-f012:**
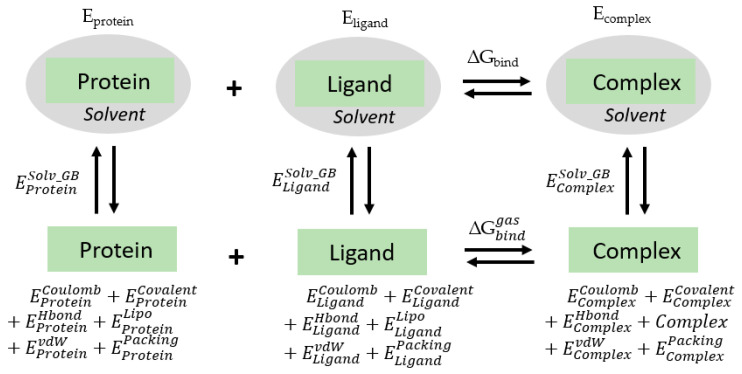
Thermodynamic cycle for binding free energy calculation.

**Table 1 ijms-22-09371-t001:** Docking scores and docking energy values for the four ER complexes.

ER Complex	Glide		QPLD	
	XP Score Kcal/mol	Docking Energy	QPLD Score Kcal/mol	Docking Energy
ERα1_E2	−11.00	−39.74	−11.45	−38.78
ERα2_E2	−9.65	−32.54	−9.82	−33.41
ERα1_OHT	−9.07	−29.43	−8.17	−28.96
ERα2_OHT	−8.59	−35.81	−10.94	−38.23

**Table 2 ijms-22-09371-t002:** Summary of the MD simulation systems.

Complex	Atoms in Complex	Waters	Ions
ERα1_E2	3980	7890	30 Na^^+^^; 23 Cl^−^
ERα1_OHT	3994	7890	29 Na^+^; 22 Cl^−^
ERα2_E2	3946	9138	36 Na^+^; 26 Cl^−^
ERα2_OHT	3960	9138	35 Na^+^; 25 Cl^−^

**Table 3 ijms-22-09371-t003:** Interaction energy values for the ER complexes.

Acronyms	ΔG_Bind	ΔG_Bind_Coulomb	ΔG_Bind_vdW	Ligand Energy	Complex Energy	Receptor Energy
ERα1_E2	−44.10	−7.73	−22.17	1.73	−9402.71	−9360.34
ERα1_OHT	−44.68	−22.03	−17.47	32.43	−9352.97	−9340.71
ERα2_E2	−24.69	−9.19	−7.13	1.72	−9325.79	−9302.81
ERα2_OHT	−42.24	−39.65	−17.04	32.12	−9342.13	−9332.01

**Table 4 ijms-22-09371-t004:** Major clusters from hierarchical clustering analysis.

ER Complex	Cluster Number	Number of Structures	Start Frame	End Frame	Representative Structure
ERα1_E2	1	4314	2	4328	2312
	2	1012	4329	5346	4545
	3	1366	5364	6787	6339
	4	7027	7423	14,514	13,922
	5	1938	14,523	16,491	15,558
	6	874	16,730	17,629	17,171
ERα1_OHT	1	1696	2	1698	979
	2	860	2456	3422	2866
	3	1328	5894	7298	7154
	4	13,528	7306	20,835	15,478
ERα2_E2	1	5247	2	5451	2353
	2	6915	5453	12,945	8216
	3	1294	15,187	16,740	16,064
	4	3786	16,735	20,835	18,646
ERα2_OHT	1	3732	5	3745	1813
	2	1531	4552	6086	5356
	3	3497	6087	9643	7771
	4	5916	9641	15,769	13,467
	5	2988	15,770	19,021	18,075

## Data Availability

Not applicable.

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
