# Peer review of "Elucidation of Agonist and Antagonist Dynamic Binding Patterns in ER-α by Integration of Molecular Docking, Molecular Dynamics Simulations and Quantum Mechanical Calculations"

_ijms, 2021, doi:10.3390/ijms22179371_

Round 1

Reviewer 1 Report

In their work, the authors present the usage of in-silico methods including molecular docking, molecular dynamics simulations with subsequent free energy calculations and quantum mechanical calculations to elucidate the binding patterns of estrogen receptor agonist 17b-estradiol and antagonist 4-hydroxytamoxifen to estrogen receptor alpha. The paper presents the results poorly and contains almost no interpretation behind the meaning of the results.  As the authors have a wealth of MD trajectories, I list below my suggestions on how more information could be obtained from said trajectories and how to improve the overall paper in terms of presentation and interpretation of results.

  • Throughout the text, the authors state that the binding mechanisms of ERa agonists and antagonists remain unclear. This is a strange statement, as by their own words, more than 350 crystal structures with various ligands exist in the PDB, from which the binding mechanisms can be observed. Maybe by binding mechanisms you mean specifically dynamic binding patterns from MD simulations (how non-covalent bonds are formed and broken throughout the simulation), and how ligands influence conformational changes which relate to receptor activation? Much more should be written about the latter.
  • The paper would be much clearer if the authors could add to the introduction a figure which would present the general structure of the ERa. This figure should contain a good visual representation of the receptor, with appropriate labeling of the three distinct domains, the twelve helices, and two b-sheets, and show the location and the structure of the binding site. Moreover, it should present the differences between the active and inactive receptor conformation.
  • On page 2, line 45, you write about AF2 binding site, and that LBD is distorted and not suitable for binding cofactors. You do not explain what is AF2, and what is the nature and importance of cofactors.
  • The importance of these two ligands should be presented in connection to their pharmacology and an overview of their chemical structure (functional groups, charge of the secondary amine on tamoxifen, etc...).
  • The presentation of the docking methods is extremely confusing. You talk about a mix of different docking and scoring methods (QPLD, Glide, QPLD scores, docking scores), with little discussion on their connection. A very clear scheme and additional discussion in the methods section of how exactly these methods are connected/follow one another should be created.
  • Figure 1 is of poor quality, and it is impossible to read the aminoacid residues. Why are there no double bonds in the ligand structures?
  • A redocking study should be performed to validate the docking procedure, specifically using the same parameters as used for your docking study: you should redock estradiol to the 1GWR binding site, whereas tamoxifen to 3ERT. You should report the RMSD between the crystal ligands and redocked ligands and present a corresponding figure. For details you can check and quote the paper Biomolecules 2021, 11(3), 479, subsection 3.1. Molecular Docking and Figure S2. Moreover, how do different docking procedures compare in terms of RMSD values? Overall, redocking is a standard procedure for docking validation.
  • In QPLD docking, which residues were part of the QM system?
  • In MD simulations, specifically page, 8, line 221, you state that MD simulations were used to remove steric clashes between atoms. If the docking was successful, appropriate poses should already be without steric clashes.
  • The MD simulations were performed using the OPLS-2005 force field, which you used for protein, water, ions as well as drug ligands. Uncritical use of general force fields for small drug-like molecules can be problematic in terms of simulation accuracy and predictive value. You should at least add a few sentences of discussion concerning this problem. What are some specific properties of this force field? For a broad discussion about force-fields for small-molecule ligands see J. Chem. Inf. Model. 2020, 60, 7, 3566–3576.
  • There is a presentation about the binding patterns in the discussion, which should be in the results section. Moreover, this should be written more systematically with added interpretation. Now you just present a list of interactions without meaning.
  • For better readability, you should add a table presenting all the MD simulations you performed.
  • What would really give added value to the paper is trying to interpret the mechanism of receptor activation. In connection to this, you should discuss your observations about conformational changes when (a) agonist is bound to the inactive receptor, and when (b) an antagonist is bound to the active one. Do you observe in (a) that the receptor’s conformation changes towards an active one, and vice versa? Moreover, by defining the appropriate “descriptor(s)” describing the change from active to inactive form (you talk about the movement of H12 away from the binding pocket, for example) you could use the standard thermodynamic relation relation ΔG = -RTln(Nactive/Ninactive), thus calculating the energies of receptor activation. For details see  Current Pharmaceutical Design, Volume 20, Number 21, 2014, pp. 3478-3487 subsection 3.3.
  • You use MM-GB/SA to calculate free energies of ligand binding. In connection to this, you just plot energies at different times in Figure 3 without any interpretation, which should urgently be added.
  • Moreover, I would strongly suggest adding an analog calculation for free energy, using the linear interaction energy (LIE). This is an established method to obtain free energies of protein-drug, protein-nucleic acid, etc., binding (e.g. Biomolecules 2020, 10(12), 1700). More importantly, LIE enables the decomposition of interaction into atom group contributions, thereby enabling the calculation of contributions to free energy of each defined ligand functional group. Quote and discuss.
  • For the free energy methods, a figure presenting the thermodynamic cycle should be presented.
  • Please discuss for MD simulations the occupancy of H-bonds (how often are they formed during simulations); which are the most stable ones, which are only transient?
  • Compare the free energy values to existing experimental measurements.

Reviewer 2 Report

The manuscript ijms-1256308 is focused on the  elucidation of the  agonist and antagonist binding mechanisms of ERα. The authors performed the molecular docking, molecular dynamics (MD) simulations, and quantum mechanical calculations.

The workflow of the study is clearly presented. However, I would like to address some comments

L133-L141 The authors underline some residues from the active site, residues relevant for the binding of the compounds analyzed. It would be useful to add in the Introduction section a brief description of the active site of ERα.

In my opinion the conclusion L273 "The elucidated binding reaction mechanisms may help discover ERα agonists and antagonists .... " is too speculative. Someone could understand that the present study elucidate the binding reaction mechanism. The present study advances the hypothesis of the agonist/antagonist conformers, but a deeper analysis is required to clarify the binding mechanism of ERα agonists and antagonists. However, the interactions of specific residues of the active site of ERα1 compared to ERα2 are of great interest for further studies. In my opinion, readers might be interested in the type of interactions of these residues, not just the hydrogen bonds mentioned in the manuscript. 

Round 2

Reviewer 1 Report

The authors successfully resolved all issues raised by this reviewer. Consequently, the manuscript has been significantly improved and can be in its current version recommended for publication in IJMS.